# The Efficacy of Vitamins in the Prevention and Treatment of Cardiovascular Disease

**DOI:** 10.3390/ijms25189761

**Published:** 2024-09-10

**Authors:** Paramjit S. Tappia, Anureet K. Shah, Naranjan S. Dhalla

**Affiliations:** 1Asper Clinical Research Institute, St. Boniface Hospital, Winnipeg, MB R2H 2A6, Canada; ptappia@sbrc.ca; 2Institute of Cardiovascular Sciences, St. Boniface Hospital Albrechtsen Research Centre, Winnipeg, MB R2H 2A6, Canada; 3Department of Nutrition and Food Science, California State University Los Angeles, Los Angeles, CA 90032, USA; akaur23@calstatela.edu; 4Department of Physiology and Pathophysiology, Max Rady College of Medicine, University of Manitoba, Winnipeg, MB R2E 0J9, Canada

**Keywords:** cardiovascular diseases, vitamin deficiency, vitamin supplements, cardiac dysfunction, cardiac arrhythmias, metabolic abnormalities, oxidative stress

## Abstract

Vitamins are known to affect the regulation of several biochemical and metabolic pathways that influence cellular function. Adequate amounts of both hydrophilic and lipophilic vitamins are required for maintaining normal cardiac and vascular function, but their deficiencies can contribute to cardiovascular abnormalities. In this regard, a deficiency in the lipophilic vitamins, such as vitamins A, D, and E, as well as in the hydrophilic vitamins, such as vitamin C and B, has been associated with suboptimal cardiovascular function, whereas additional intakes have been suggested to reduce the risk of atherosclerosis, hypertension, ischemic heart disease, arrhythmias, and heart failure. Here, we have attempted to describe the association between low vitamin status and cardiovascular disease, and to offer a discussion on the efficacy of vitamins. While there are inconsistencies in the impact of a deficiency in vitamins on the development of cardiovascular disease and the benefits associated with supplementation, this review proposes that specific vitamins may contribute to the prevention of cardiovascular disease in individuals at risk rather than serve as an adjunct therapy.

## 1. Introduction

There are 13 vitamins that are essential for human health, and they are grouped according to whether they are fat-soluble (lipophilic) or water-soluble (hydrophilic). The lipophilic vitamins are designated by the letters A, D, E, and K and extensive information exists on their diverse cellular and metabolic actions. For example, traditionally, vitamin K is known to be required for normal blood-clotting processes, vitamin A (retinol) is the precursor of retinal, the light-absorbing group in visual pigments, vitamin D is a key regulator of calcium and phosphorus metabolism, and vitamin E is a potent antioxidant that protects membrane lipid peroxidation. On the other hand, the hydrophilic vitamins are the B vitamin series and vitamin C. With respect to their functionality, vitamin C is a potent reducing agent, whereas B vitamins are the components of cofactors—for example, thiamin (vitamin B1) is a precursor of thiamin pyrophosphate, whereas riboflavin (vitamin B2) is a precursor of flavin adenine dinucleotide—or coenzymes—for example, pantothenate (vitamin B5) is a component of coenzyme A. Indeed, since both lipophilic and hydrophilic vitamins are involved in diverse biochemical and molecular processes and cell function, they are considered as essential micronutrients for optimal health. In view of the widespread availability and consumption of foods that are naturally enriched with vitamins, a deficiency is not usually observed in healthy individuals.

According to Statistics Canada, in the first half of 2012, the sales of vitamins and other health supplements were approximately $1.7 billion CDN and, in the USA, $30 billion US is spent on dietary supplements, including vitamins, per year. Moreover, Fortune Business Insights has estimated that the vitamins and supplements global market size was approximately $120 billion US. In view of the ever-increasing interest and concern for health and wellbeing, consumers are increasing their efforts to incorporate vitamins into their diet. Indeed, it has been estimated that 60% of consumers worldwide are taking vitamin supplements as part of their daily routine. Most individuals appear to consume vitamin supplements for the prevention of cardiovascular disease (CVD) [1] based on health claims, for example, the B vitamins can increase energy production and improve cardiovascular health. Vitamin C, as an antioxidant, can reduce endothelial (the cell lining of blood vessels) oxidative damage. Vitamin E not only serves as an antioxidant but can also regulate platelet function and inhibit the formation of foam cells, while insufficient levels of vitamin D may increase the risk of CVD. These claims have led to extensive research in the area of vitamins and cardiovascular health, as well as public consumption and expenditure.

We have recently edited books on the functional aspects of both lipophilic and hydrophilic vitamins in health and in various pathophysiological conditions [2,3]. While plasma vitamin levels are maintained within normal defined ranges in healthy individuals, deficiencies in both lipophilic and hydrophilic vitamins have been reported in different cardiac pathologies [4,5,6,7,8,9,10]. Two major factors are implicated in the pathogenesis of CVD: oxidative stress and increased intracellular Ca^2+^ concentration ([Ca^2+^]_i_) or Ca^2+^ overload [11,12,13,14]. There is a significant amount of experimental evidence that vitamins may not only influence these processes but may also regulate other contributors in the pathophysiology of CVD, including the dysregulation of lipid metabolism, increases in homocysteine levels (a marker for CVD), nitric oxide concentrations (a gaseous vasodilator molecule), immune-modulatory effects, and the activation of inflammatory processes [15,16,17,18,19,20]. Thus, specific vitamins have been recommended in the promotion of cardiovascular health [21,22,23,24,25,26,27,28]. Since the etiology of CVD is multifaceted, the efficacy, as well as the identification of mechanistic targets, may be different under optimal health conditions vs. pathophysiological conditions. Figure 1 summarizes the different cardiac and vascular outcomes that could potentially occur during specific hydrophilic vitamin deficiencies, whereas Figure 2 depicts the potential impact of lipophilic vitamin deficiency and the development of different cardiovascular disorders. 

Public awareness and scientific interest in the influence of vitamins on cardiovascular health have increased exponentially. Therefore, in this article, we discuss the efficacy of different vitamins as part of a nutritional approach in the prevention of CVD and as potential therapeutic agents in the treatment of CVD. It is also intended to examine if there is a relationship between deficiencies in specific vitamins and the etiologies of specific types of cardiovascular pathologies. In view of the role of oxidative stress on the pathogenesis of CVD, and given that different vitamins exhibit antioxidant properties [29,30,31,32,33,34], the effects of different vitamins will also be discussed with respect to their antioxidant capacity to attenuate oxidative stress as well as their ability to exert anti-inflammatory actions in CVD. Accordingly, this review was generated through the collection of appropriate literature searched on MEDLINE via PubMed by using the search terms: cardiovascular disease and vitamins, and individual vitamins in relation to heart disease/cardiovascular disease and heart failure, and combinations thereof. 

## 2. Vitamin B Deficiency and Cardiovascular Disease

A depletion in energy stores has been observed in patients with heart failure that has been attributed to a deficiency in vitamins B1, B2, and B6 [35]. Vitamin B1 deficiency has also been linked to CVD risk factors, such as diabetes, dyslipidemia, obesity, and vascular inflammation [8], while a deficiency in vitamin B2 in CVD has been associated with anemia and increased homocysteine concentrations [10,36]. Furthermore, increases in homocysteine levels have also been reported in patients with coronary artery and peripheral artery diseases with deficiencies in both vitamins B6 and B12 [37,38]. In addition, vitamin B6 deficiency has been implicated in some vascular disorders, including hypertension, atherosclerosis (hardening of arteries under oxidative stress), and coronary artery disease (CAD), a narrowing or blockage of the coronary arteries [39,40,41,42]. 

Interestingly, thiamin has been reported to improve mortality rates in an experimental model of cardiac arrest in the mouse and thus thiamin has been proposed to improve cardiovascular outcomes in patients post cardiac arrest [43]. Syed et al. [44] have recently conducted a systematic review and meta-analysis of six randomized controlled trials (RCTs) involving a total of 298 participants with a diagnosis of heart failure treated with thiamin. No differences in the left ventricle ejection fraction (LVEF), N-terminal pro–B-type natriuretic peptide (NT-proBNP), a diagnostic marker for heart failure, and left ventricle end-diastolic pressure were found between the intervention group and placebo; however, thiamin supplementation was able to improve the heart rate [44]. Another meta-analysis involving seven RCTs and 274 patients with chronic heart failure revealed that thiamin supplementation did not improve the function of the failing heart [45]. On the other hand, better outcomes in patients with heart failure have been reported following a long-term supplementation with thiamin, as evidenced by the increase in time to a cardiac event leading to death [46]. Similarly, a systematic review and meta-analysis of eight studies involving 384 patients with heart failure showed that supplementation with thiamin was not associated with an improvement in LVEF. Interestingly, thiamin status was unaffected by thiamin supplementation [47]. In view of the variations in the results with thiamin supplementation, it has been suggested that while thiamin supplementation may improve heart function in heart failure due to thiamin deficiency, it may be of no benefit in heart failure in general [48].

Experimental investigations have reported the occurrence of atherosclerosis [49], CAD [50], elevated blood pressure [51], and increased sympathetic nervous activity [52] in rats fed a diet deficient in vitamin B6. In addition, while defects in the regulation of cardiomyocytes [Ca^2+^]_i_, as well as sarcolemmal ATP receptor function in vitamin B6-deficient rats have been observed, these alterations were completely reversed, suggesting that myocardial abnormalities may be related to vitamin B6 deficiency [53]. Suboptimal levels of vitamin B12 are linked to abnormal endothelial function in diabetes, atherosclerosis, myocardial infarction (MI) or heart attack, and stroke [7,54,55]. Vascular disorders, as well as congenital heart disease (structural defects of the heart that are present at birth), have been associated with diminished vitamin B9 (Folic acid) levels [4,20,56]. 

It is interesting to note that high intakes of vitamin B2 (riboflavin), B6, B9, and B12 have been linked to a reduced risk of hypertension and induce a lowering of blood pressure in hypertensive individuals [57,58,59]. A complex of vitamins B1 (thiamin), B2, B6, B9, and B12 has also been reported to be effective in the management of heart failure [34,60,61]. A combination of vitamins B9 and B12 has been shown to attenuate myocardial cell damage induced by isoproterenol, reduce homocysteine levels, and diminish oxidative stress in hyperhomocysteinemic rats (an experimental model of abnormally high homocysteine levels) [62]. Folate can reverse both endothelial dysfunction and the increased production of superoxide anions induced by the depletion in tetrahydrobiopterin (BH4) levels with 2,4-diamino-6-hydroxy-pyrimidine and N-acetyl-5-hydroxy-tryptamine I in the rabbit aortic ring [63]. Indeed, the supplementation of folate was reported to improve the impairment of endothelial function in CVD [20]. Interestingly, vitamins B6 and B12, along with folate, have been demonstrated to have an inverse relationship with hypertension [64].

Niacin treatment in patients with hypertriglyceridemia can reduce total cholesterol levels and triglycerides and thus it was suggested that this B vitamin may reduce cardiovascular outcomes [65]. A B vitamin complex of vitamins B1, B2, B6, B12, and folic acid was found to reduce atherosclerosis and ischemic heart disease (IHD), a condition where the heart is deprived of oxygen due to a reduced blood supply, by virtue of their anti-inflammatory actions [66]. Both vitamins B9 and B12 can delay the early onset of CAD by reducing plasma homocysteine levels [67]. In addition, a reduction in the risk of endothelial dysfunction in patients with CAD has been reported to occur following treatment with folate and other vitamins [68,69]. The co-administration of folic acid, and vitamins B6 and B12 in patients with hyperhomocysteinemia was observed to reverse endothelial dysfunction [70]. While these lines of evidence indicate favorable outcomes of specific B vitamins in CVD, a meta-analysis has shown that vitamins B6, B9, and B12 do not protect against the progression of atherosclerosis [71].

The cardiovascular outcomes in a UK population study with 115,664 participants have been evaluated with regard to B vitamin dietary intakes [72]. It was revealed that with each increment of vitamin B9 (folate) intake, the risk of total CVD events was reduced by 5%, whereas deaths due to CVD were lowered by 10% [72]. It was thus proposed that folate intake might be of benefit as part of a strategy for the primary prevention of CVD. Since elevated homocysteine levels are considered a strong marker for CVD and vitamin B6 is an important component in the regulation of homocysteine metabolism, several experimental studies (reviewed in [73]) have revealed the potential of vitamin B6 through antioxidative and anti-inflammatory actions [73,74], as well as by exerting a vasodilatory effect and improving coronary flow [73]. Indeed, homocysteine-lowering B vitamins could be part of a strategy for the prevention of CVD [75]. Vitamin B6 is an important molecule that is involved in metabolism and cell signal transduction [76]. It can also exert antioxidant effects and reduce advanced glycation end products (glycated protein and lipid biomarkers for different diseases, including atherosclerosis).

The potential of pyridoxal 5′-phosphate (PLP), a product of vitamin B6 metabolism, in the treatment of IHD has been explored [29,77,78,79,80]. In this regard, PLP has been shown to inhibit reactive oxygen species formation and the oxidation of membrane lipids subsequent to H_2_O_2_ production [81]. Furthermore, PLP is able to diminish the ATP-induced increase in cardiomyocytes [Ca^2+^]_i_ and the binding of ATP to the sarcolemma membrane [82]. Not only can PLP reduce cardiac dysfunction due to I/R, it can also diminish infarct size [79,83]. Clinical observations have revealed that PLP can attenuate ischemic injury following different coronary surgical procedures [84,85]. However, in a subsequent large clinical trial, PLP was found not to exert any beneficial effect in high-risk patients undergoing coronary artery bypass graft surgery [86,87]. While the ineffectiveness of PLP in attenuating adverse cardiovascular effects in patients with IHD is perplexing, the pretreatment of experimental rats with PLP attenuates MI-induced arrhythmias, ventricular tachycardia, and mortality [80]. In addition, I/R-induced cardiac dysfunction, as well as changes in sarcoplasmic reticulum Ca^2+^-uptake and Ca^2+^-release, were prevented by the pretreatment of rats with PLP [80]. Taken together, it is conceivable that PLP is cardioprotective rather than a therapy for IHD. Plasma PLP has been reported to be low in inflammatory conditions [88]. An anti-inflammatory action of vitamin B6 has been attributed to an immunomodulatory effect of vitamin B6 [64]. Overall, while vitamin B6 can modulate [Ca^2+^]_i_, on its own, it appears that B vitamins together as a complex may be more effective in attenuating cardiovascular abnormalities and/or diminishing the risk of CVD. A B vitamin complex would be seen to exert preventive actions and maintain optimal cardiovascular health and function. On this basis, B vitamins (B1, B2, B6, B9, and B12) could be recommended as part of a strategy for the prevention of CVD.

## 3. Vitamin C and Cardiovascular Disease

Although low levels of plasma vitamin C due to its decreased intake have been reported to be associated with a high risk of cardiovascular disease [89,90], the relationship between the plasma levels of vitamin C and the risk for cardiovascular events is not clear at present. The risk of coronary artery disease in women was found to increase due to vitamin C deficiency [91], probably as a consequence of the increased oxidation of low-density lipoproteins and the development of atherosclerosis [92,93]. On the other hand, vitamin C deficiency in elderly people was related to the risk of death from stroke rather than from coronary artery disease [94]. Nevertheless, a deficiency in vitamin C has been linked to an increase in the risk of acute MI in men [95]. Interestingly, an inverse relationship between the high sensitivity of C-reactive protein and vitamin C has been reported in heart failure [9]. In addition, because vitamin C is a potent antioxidant [18,96], plasma concentrations of this vitamin appear to be predictive of heart failure [97]. 

Population studies in IHD have revealed a reduction in atherosclerosis by vitamin C that has been attributed to an attenuation of endothelial dysfunction and an improvement in blood lipid concentrations, as well as by preventing the formation of oxidized LDL [98]. A reduction in the risk of IHD has also been reported in heavy smokers with a higher consumption of vitamin C [99]. In fact, the treatment of patients with IHD with vitamin C has been shown to re-establish coronary flow and inhibit the re-induction of coronary constriction [100]. Furthermore, vitamin C is cardioprotective against I/R-induced oxidative damage [101]. Experimental studies in mice have revealed that the beneficial effects of vitamin C in IHD are attributed to improving hyperlipidemia, as well as HDL remodeling [102]. While vitamin C appears to be protective in IHD, there are some studies that are not conclusive for a favorable impact of vitamin C in other types of CVD. For example, although blood pressure reduction has been reported in patients with hypertension following the administration of vitamin C [103,104], a similar response to vitamin C in pregnancy-induced hypertension has not been observed [105]. Likewise, while the risk of postoperative atrial fibrillation can be reduced by vitamin C [106], it has no effect on atrial fibrillation due to atrial-tachycardia remodeling in dogs [107]. Similarly, vitamin C exerted no beneficial actions on endothelial dysfunction and no improvement in atherosclerosis [108,109]. Thus, from the aforementioned, it would appear that vitamin C may be of benefit in the treatment of IHD, but not in other types of CVD. From the aforementioned, it is evident that vitamin C is a potent antioxidant vitamin that has a greater potential for preventing CVD under conditions of oxidative stress, specifically in IHD through antioxidative mechanisms. A summary of some of the major mechanisms of action of the hydrophilic vitamins are depicted in Figure 3.

## 4. Vitamin A and Cardiovascular Disease

Vitamin A and its precursors, α-carotene and β-carotene, have been claimed to exert beneficial effects on the development of different cardiovascular diseases [110,111,112]. In fact, vitamin A and β-carotene may protect against lipid peroxidation [113]. Vitamin A treatment in individuals with elevated blood pressure has been shown to reduce the systolic and diastolic blood pressures [114]. Interestingly, the sustained use of vitamin A has been reported to decrease the atherosclerotic process in both animals and in humans as a result of its antioxidant and anti-inflammatory properties [115,116]. Similarly, in patients with diabetes with IHD, the extent of oxidative stress has been reported to be lowered subsequent to treatment with vitamin A [117]. The β-carotene treatment of Zucker diabetic rats has been shown to reduce the infarct size subsequent to I/R and improve the recovery of cardiac function post ischemia [118]. It is noteworthy that treatment with β-carotene was cardioprotective against advance glycation end product-induced SR stress and cell death, as well as autophagy [119]. On the other hand, β-carotene has been shown to exert no beneficial actions on metabolic syndrome induced by the high-fat feeding of rats [120].

Vitamin A (retinol) is increasingly gaining attention as an important lipophilic vitamin for optimal cardiometabolic health [121]. While experimental studies have shown that vitamin A can modulate different biological processes, including immune response and lipid metabolism, conclusive clinical evidence is lacking [122,123]. Some human studies [123] have shown that retinol was inversely and positively linked to CVD, while retinoic acid (a metabolite of vitamin A) was found to be negatively correlated with CVD [123]. Retinoid (the active form of vitamin A)-mediated signaling has been suggested to be involved in the pathological remodeling of the vasculature, as well as of the heart [124]. It should be mentioned that since vitamin A is important in heart morphogenesis during in utero development [125], it has thus been suggested that the regulation of genes involved in heart development that are influenced by both a deficiency or an excess of vitamin A may have a critical role in inducing congenital heart disease [126]. It has also been suggested that vitamin A may prevent restenosis through anti-inflammatory and immunomodulatory actions [127]. Despite the reported antioxidant and anti-inflammatory actions of vitamin A, it is less well explored for its potential in CVD. While further investigation is warranted, a recommendation for the supplemental use of vitamin A in the diet for the prevention or treatment of CVD is not permitted at this time.

## 5. Vitamin D and Cardiovascular Disease

Since vitamin D deficiency is the most common nutritional problem, extensive research efforts have been made to understand its relationship with cardiovascular disorders, as well as the mechanisms of its impact on cardiovascular function. Several investigators have emphasized that vitamin D deficiency plays a critical role in the pathogenesis of cardiovascular disease, including hypertension, heart failure, and IHD [128,129,130,131,132]. It is noteworthy that congestive heart failure in vitamin D deficiency was associated with impaired systolic and diastolic functions, hypertension, and peripheral vascular disease [129]. The development of atherosclerosis and the risk of MI have been linked to sub-optimal plasma concentrations of vitamin D, which have been attributed to an increased vulnerability to the occurrence of impaired endothelial function and autoimmune and inflammatory reactions, as well as increased foam cell formation and smooth muscle cell proliferation [19,133,134].

Several pathophysiological conditions, including dyslipidemia, metabolic syndrome, obesity, diabetes, and hypertension, have been linked to being in a vitamin D-deficient state [135,136,137]. The occurrence of low vitamin D in diabetes, as well as in hypertension, has been suggested to result in insulin resistance, an increase in parathyroid hormone levels, activation of the renin–angiotensin system, and dysregulation of nitric oxide formation, as well as oxidative and inflammatory stress [138,139,140]. Indeed, vitamin D has been implicated in the activation of the renin–angiotensin system, cardiac hypertrophy, and arterial hypertension in vitamin D receptor knockout mice [6]. An attenuation of the renin–angiotensin system and the secretion of parathyroid hormone (PTH) have been observed in individuals with elevated blood pressure, as well as heart failure, following treatment with vitamin D (calcitriol) [141,142]. 

A meta-analysis conducted of several observational studies has shown an inverse relationship between 25-hydroxyvitamin D plasma concentrations and lowered incidences of MI, heart failure, and aortic stenosis [143,144]. In addition, vitamin D treatment was revealed to slow down the progression of coronary artery disease and the development of acute MI due to the deactivation of the NF-kB pathway [145]. An attenuation in the development of atherosclerosis has been attributed to reductions in serum total cholesterol, triglycerides, and LDL, and increased HDL and endothelial nitric oxide production following treatment with 25-hydroxyvitamin D [146]. Vitamin D supplementation in high fat-fed obese rats has been reported to reduce oxidative stress and inflammation [16]. However, some clinical studies have not been able to demonstrate a protective role of vitamin D in preventing IHD or reducing mortality [24,147,148]. Similarly, the findings of RCTs in chronic heart failure and other CVDs involving vitamin D treatment have been inconclusive and contrary [149,150,151]. 

Sonderman et al. [152] have suggested concentrations > 20 ng/mL (50 nM) of 25-OH vitamin D as optimal, while 25-OH vitamin D concentrations of 40–60 ng/mL (100–150 nM) are considered as sufficient according to Mason et al. [153]. Such variability in the definition of optimal/sufficient vitamin D may be a strong determinant in the interpretation, as well as in the effectiveness, of vitamin D. While a deficiency in vitamin D has been associated with different cardiac and vascular pathophysiological conditions [154], the findings of some vitamin D interventional trials with individuals at risk or with CVD are, however, inconclusive [149,150,151,155]. 

Several studies have reported positive cardiovascular outcomes with supplementation with vitamin D. In this regard, the EVITA trial (Effect of Vitamin D on mortality in heart failure) with a 4000 IU vitamin D supplementation in individuals with advanced stages of heart failure every day for up to 36 months showed some benefit on LV ejection fraction, but only in those patients who were ≥50 years of age [156]. Similarly, another study involving 4000 IU vitamin D/day for 6 months also revealed an improvement in the LV ejection fraction in elderly (mean age of 74 years) individuals with a failing heart. In addition, the blood pressure was reduced in this group. It should be mentioned that all the study participants were determined to have a deficiency of vitamin D with serum concentrations of <30 ng/mL before the initiation of vitamin D treatment in the study. Importantly, a marked increase in circulating levels of 25-OH vitamin D was measured in these individuals following supplementation [157]. Interestingly, an increased risk in mortality in patients with advanced stage/chronic heart failure has been inversely linked to low 25-OH vitamin D in the blood [158,159]. However, in contrast, the EVITA trial revealed that even though vitamin D treatment restored optimal vitamin D concentrations, the mortality rate was not reduced in those subjects with advanced heart failure. 

In another RCT involving patients with Class II-III heart failure who were deficient in 25-OH vitamin D (<32 ng/mL), 10,000 IU vitamin D3 daily for 6 months was observed to correct the 25-OH vitamin D-deficient state and improve quality of life measures [160]. Furthermore, decreases in B-type natriuretic peptide, PTH, and high-sensitivity C-reactive protein levels were observed in the treatment group [161]. Conversely, in another RCT involving patients with heart failure, with an average age of 65.9 years and an average ejection fraction of 37.6%, receiving 50,000 IU vitamin D3/week for 6 months, no improvements in physical performances were observed. In addition, no changes in the peak VO2 (a measure of peak oxygen uptake in inspired air), 6-min walk test (6-MWT), and knee isokinetic muscle strength were observed, even though the serum 25-OH vitamin D concentrations were increased [161]. Interestingly, 48% of the study participants were women and 64% were African American. This presents an intriguing question on the influence of sex and ethnicity on the efficacy of vitamin D treatment, although vitamin D was shown to be ineffective in decreasing risk factors for CVD (i.e., hypertension and abnormal lipid profile) in vitamin D-deficient post-menopausal women [162].

The Effects of Cholecalciferol Supplementation in Patients with stable heart failure and LOw vITamin D levels (ECSPLOIT-D) study conducted in individuals with stable heart failure and a serum vitamin D concentration of <20 ng/mL reported that an initial loading dose of 300,000 U oral cholecalciferol, and then 50,000 U cholecalciferol/month for 6 months, improved function as evidenced by a better 6-MWT, but this was observed only at 3 months of the supplementation period [163]. On the other hand, the VINDICATE heart failure study found that a daily supplementation with 4000 IU vitamin D for 12 months did not improve the 6-MWT outcome in those participants that exhibited initial serum vitamin D concentrations of <20 ng/mL [164]. However, the intervention with vitamin D increased LVEF and reduced the LV end-diastolic diameter and the LV end-systolic diameter. These investigators suggested that vitamin D may reverse the remodeling of the LV and augment cardiac contractile function in heart failure [164]. It should be mentioned that subsequent to a secondary analysis of the data in the EVITA study, vitamin D supplementation was found not to improve the blood lipid profile. Furthermore, no differences in the levels of the vascular calcification inhibitor, fetuin-A, were observed [165]. Accordingly, an intervention with vitamin D does not lower CVD risk in this study population. 

The existence of anemia is often associated with advanced stage heart failure that has been linked to low circulating levels of 25-OH vitamin Ds [166]; however, a daily supplementation of 4000 IU of vitamin D for 3 years was reported not to reduce anemia. An increase in the risk of worsening function of the kidney in chronic heart failure has been correlated to a diminished 1,25-dihydroxyvitamin D/parathyroid hormone ratio [167]. In fact, this ratio was recognized as an independent risk factor for adverse cardiovascular events that required hospitalization and for mortality [167]. It is noteworthy that the degree of deficiency of vitamin D is connected to in-hospital cardiac events in individuals admitted to hospital following the first acute MI [168]. Conversely, the EVITA heart failure study reported that vitamin D intervention did not lower the rate of first heart failure hospitalizations [169]. 

Since there is an inconsistency in the demonstration that vitamin D exerts beneficial effects, it is, therefore, plausible that maintaining optimal concentrations of vitamin D levels may prevent the development of cardiac disease. Increases in the heart rate, as well as in systolic and diastolic blood pressures, have been reported in individuals deemed as vitamin D deficient [170]. These observations led to the suggestion that this vitamin might modulate the sympathetic nervous system, as well as influence norepinephrine levels. In addition, since these parameters were not affected subsequent to long-term vitamin D supplementation, it is plausible that vitamin D may prevent heart disease as long as optimal levels are preserved.

It should be mentioned that the optimal vitamin D concentration assumes that there is no influence of ethnicity on the requirement for vitamin D. In this context, a study has shown that in order to achieve optimal vitamin D levels, Somali women have to ingest more than twice the amount consumed by Finnish women [171]. On the other hand, inadequate vitamin D levels among women of East African descent residing in Finland has been reported, despite the higher vitamin D intakes [172]. It should be mentioned that the National Health and Nutrition Examination Survey of children in 2006 revealed that <1% of non-Hispanic Black children exhibited optimal vitamin D concentrations, which was markedly lower than the 25% of non-Hispanic White children who had adequate vitamin D levels [173]. Taken together, it was thus suggested that ethnicity is a more important determinant in the recommendation of vitamin D supplementation than seasonal or latitude variations. Nonetheless, despite >50% of the population of the world estimated as being insufficient in vitamin D, any approach undertaken to prevent disease, including cardiac and vascular abnormalities, must include vitamin D supplementation, particularly where there is a risk for a lower and sustained exposure to sunlight, as well as increasing the consumption of seafood and other food sources of vitamin D, such as liver and fortified foods [174]. 

The uncertainties around vitamin D have led to a discord in the recommendation of vitamin D supplementation in the treatment of CVD or as a preventive agent in patients deemed to be at risk of CVD. We have recently described the challenges in the interpretation and understanding of the clinical relevance of vitamin D in relation to cardiovascular health and function [175], thus more research is required. From the aforementioned, however, it is possible that maintaining the optimal concentration of vitamin D may be cardioprotective and prevent CVD. 

## 6. Vitamin E and Cardiovascular Disease

Since vitamin E is widely consumed through food, its deficiency is rarely seen [21,23]. However, there are vulnerable populations, such as infants, individuals with fat malabsorption, or individuals with some genetic conditions, where vitamin E deficiency can occur [176,177]. There are several lines of both clinical and experimental evidence that support the efficacy of vitamin E in relation to cardiovascular health. For example, a lowering of blood pressure in patients with essential hypertension subsequent to vitamin E supplementation has been reported [103]. In addition, supplementation with vitamin E can prevent complications due to a pregnancy-related elevation in blood pressure [105]. Treatment with vitamin E has also been reported to slow the rate of atherosclerotic disease progression and reduce endothelial dysfunction [109,178]. In fact, a lower risk of CAD in men has been seen in response to vitamin E treatment [17]. Experimentally, vitamin E has been reported to prevent ischemic heart disease and ischemia-reperfusion injury [102,179,180]. In addition, positive effects of vitamin E have been observed in MI [181,182], ventricular arrhythmias due to MI [183], and arrhythmias due to the excessive production of catecholamines [184,185,186] as well as in catecholamine-induced cardiomyopathy [187]. These effects of vitamin E were attributed to reduced oxidative stress, lipid peroxidation, myocardial cell damage, and subcellular abnormalities.

In contrast, several RCTs involving supplementation with vitamin E have not evidenced any beneficial outcomes with respect to the prevention of CHD, which could be related to several factors, including differences in the coronary risks of the study patients [188], and in the design and quality of the study, as well as other limitations, including dosing, the treatment regimen, the type of vitamin E supplement, and the selection of study patients [189,190,191,192]. Despite these concerns, a low dose of vitamin E can reduce the risk of angina in individuals who do not have a history of CAD, whereas a higher dose can lower MI and cardiovascular mortality [193]. However, some caution must be exercised as high vitamin E doses have been reported to increase CAD and MI risk [194]. Discrepancies in the effects of vitamin E in atherosclerosis and coronary artery calcification have been observed [109,195]. Nonetheless, in addition to the antioxidant effects of vitamin E, the reduction in tissue damage, and the preservation of myocardial viability are indicated to provide the beneficial effects of vitamin E in MI [182]. The antioxidant and anti-inflammatory actions of α-tocopherol have resulted in a reduction in infarct size, as well as the restoration of cardiac function in the I/R of the mouse heart [180]. It is important to note that the anti-inflammatory effects of vitamin E seem to be related to the dose and the type of vitamin E supplement, i.e., α- or β-tocopherol [196]. 

It is interesting that vitamin E supplementation has no effect in post-menopausal women with CHD [22]. In contrast, vitamin E is considered to lower the risk of CHD in middle-aged to older women and in men but provides no benefit in primary and secondary cardiovascular events [5]. Indeed, while vitamin E is ineffective in diminishing cardiovascular events in high-risk patients [25], it may be protective in those patients with CHD with no CVD [23]. The Women’s Health Study of 39,876 healthy women who were at least 45 years of age revealed that vitamin E is not helpful in major cardiovascular events and cardiovascular mortality [197], it was thus proposed that vitamin E may not be effective in the prevention of CVD in healthy women. 

The US Preventive Services Task Force (USPSTF) investigated micronutrient efficacy in lowering CVD risk, as well as in lowering the mortality rate in the general population [198]. In the pooled analysis involving 284 studies and a total of 739,803 participants, it was determined that supplemental vitamin E does not prevent CVD [199]. In addition, in a meta-analysis of 884 RCTs with a total of 883,627 participants where the impact on CVD of 27 different micronutrients, including vitamin E, was evaluated, it was found that vitamin E was of no benefit in CVD [199]. Furthermore, a review of the Framingham Heart Study discovered that the extent of established CVD had no bearing on the relationship between vitamin E and CVD risk and all-cause mortality. It was thus suggested that more studies evaluating the influence of vitamin E need to be undertaken [200].

The initial Heart Outcomes Prevention Evaluation (HOPE) trial, which was later extended to the HOPE-The Ongoing Outcomes (TOO) trial [201] examined the outcomes of long-term vitamin E on the incidence of cardiovascular events in patients with diabetes and in individuals with vascular disease. Both trials involving 13,535 participants showed that the prolonged administration of vitamin E did not prevent major cardiovascular events under these pathophysiological conditions; of concern was that the risk for heart failure was actually increased [201]. Similarly, another meta-analysis of 84 RCTs of vitamin E showed that vitamin E alone or in combination with other agents was not beneficial and was not associated with all-cause and cardiovascular mortality [202]. The ineffectiveness of vitamin E could be explained on the basis that these study participants may already be getting adequate amounts of vitamin E from their daily food intake. Furthermore, in view of vascular inflammation, which is considered as a major factor in CVD, additional dietary antioxidants, including vitamin E, may be of benefit [203]. It should be mentioned that the primary reason for death in non-alcoholic fatty liver disease (NAFLD) is CVD. Preclinical investigations have shown that ornithine aspartate in combination with vitamin E can attenuate lipid metabolic disorders and improve endothelial function, which could have therapeutic implications in CVD associated with NAFLD [204]. Furthermore, although daily supplemental vitamin E is recommended in individuals with end-stage renal disease undergoing hemodialysis for preventing the occurrence of cardiovascular events, it has also been proposed to be preventive of cardiovascular events in all patients with uremia [205]. From the aforementioned, it appears that there is no conclusive evidence that taking vitamin E supplements can reduce the risk of heart disease; however, in NAFLD and in patients undergoing hemodialysis for ESRD, vitamin E may be of benefit in preventing subsequent cardiovascular events. Since high levels of vitamin E can be associated with adverse cardiovascular outcomes, some caution must be exercised in the recommendation of supplementation with vitamin E in CVD risk prevention. A summary of some of the major mechanisms of action of the lipophilic vitamins is represented in Figure 4.

## 7. Perspectives

In this review, we have evaluated the benefits and adverse outcomes of some lipophilic and hydrophilic vitamins in different cardiovascular pathologies. We have described some of the experimental and clinical lines of evidence that support the role of specific vitamin deficiencies in the development of CVD and some lines of evidence that have observed no beneficial actions of vitamins. One reason for this variance could be that in many clinical studies examining the impact of vitamins in CVD, baseline blood vitamin concentrations prior to initiating the intervention have not been measured. It is therefore conceivable that the positive outcomes of vitamin interventions are only observed in patients with insufficient/deficient blood vitamin concentrations at baseline and thus could explain why favorable outcomes of some large RCTs with vitamins have been unsuccessful. Therefore, the efficacy of vitamins should be investigated in those study cohorts that are established to be in a deficient state. In this regard, difficulties in assessing vitamin status can present a challenge. 

Extensive investigations are required for understanding the response to vitamins as a function of the dose, as well as the time period of the intervention with the different lipophilic and hydrophilic vitamins in CVD. Indeed, specific patient populations might benefit from supplementation with specific vitamins. Since there appears to be a widespread and unnecessary consumption of vitamins worldwide, it may be prudent to limit use and recommendations to those vitamins for which there is strong and credible science.

## 8. Conclusions

Due to inconsistencies between the experimental and clinical observations regarding the efficacy of vitamins in the treatment of CVD, no firm conclusions can be drawn, and it is important to explore the causes for discord. While additional exploration is warranted, new therapeutic approaches involving vitamins could be developed as adjuncts for a specific type of CVD; however, it is more likely that positive outcomes with vitamins may be in the prevention of cardiovascular aberrations of different etiologies. 

## Figures and Tables

**Figure 1 ijms-25-09761-f001:**
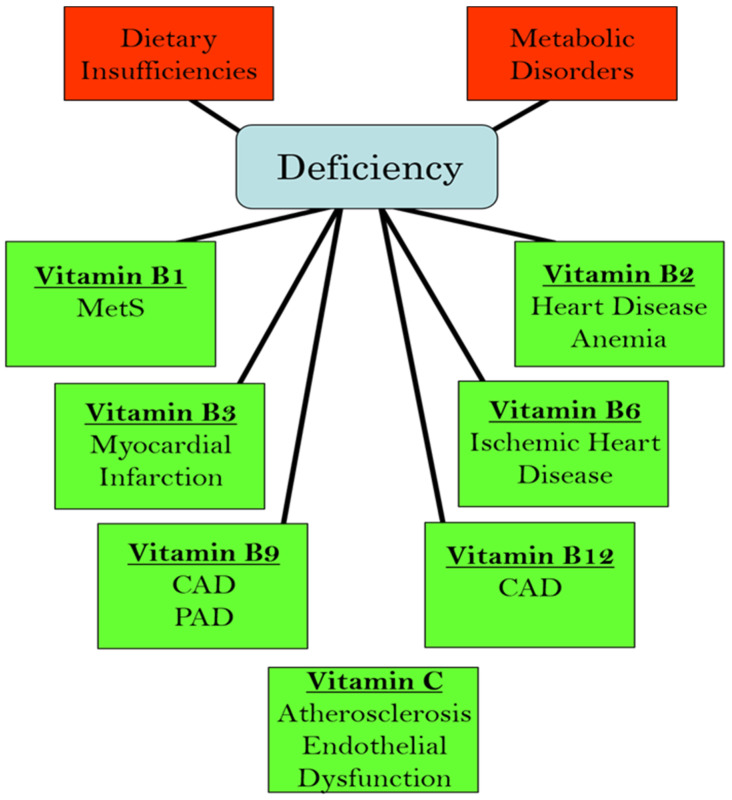
Relationship between deficiencies in different hydrophilic (water-soluble) vitamins and the development of various cardiovascular pathologies. Abbreviations: MetS, metabolic syndrome; CAD, coronary artery disease; PAD, peripheral arterial disease.

**Figure 2 ijms-25-09761-f002:**
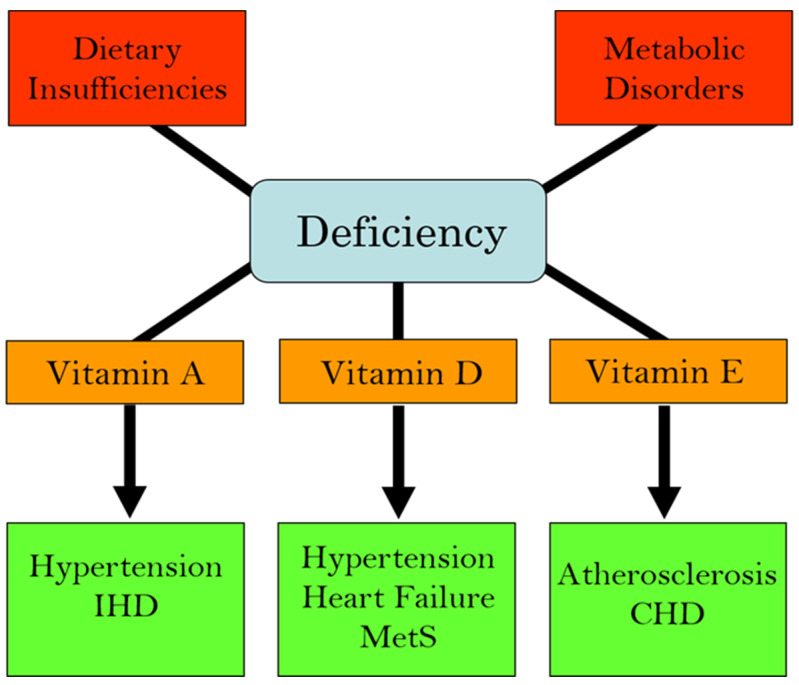
Association of deficiencies in lipophilic (fat-soluble) vitamins with different cardiovascular diseases. Abbreviations: IHD, ischemic heart disease; MetS, metabolic syndrome; CHD, coronary heart disease.

**Figure 3 ijms-25-09761-f003:**
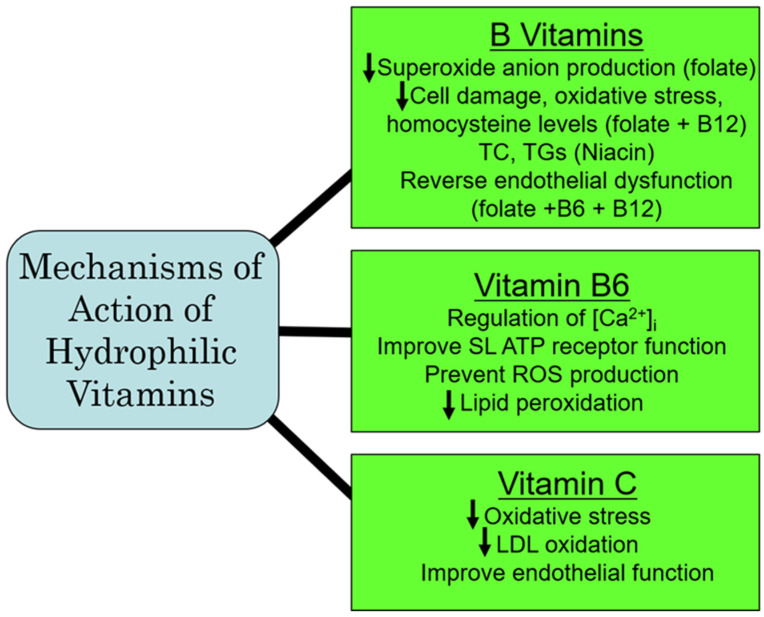
Proposed major mechanisms of action of some hydrophilic vitamins. Abbreviations: TC, total cholesterol; TG, triglyceride; [Ca^2+^]_i_, cardiomyocyte intracellular calcium concentration; SL, sarcolemmal membrane; ROS, reactive oxygen species; LDL, low-density lipoprotein.

**Figure 4 ijms-25-09761-f004:**
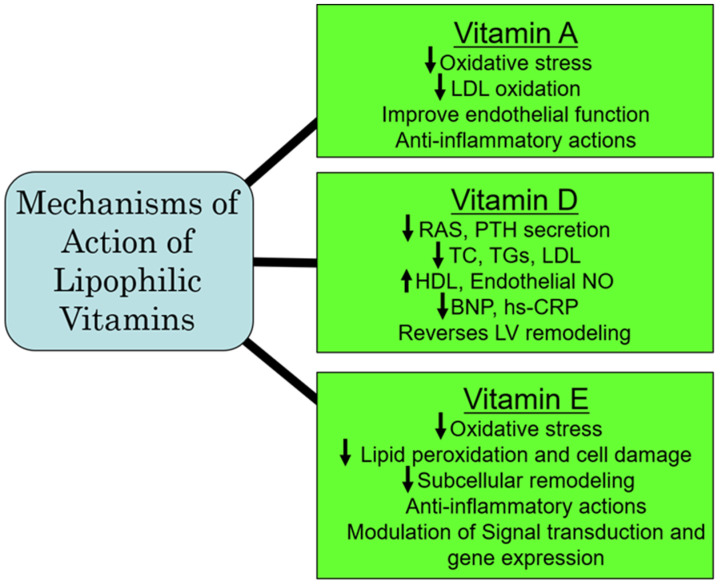
Proposed major mechanisms of action of some lipophilic vitamins. Abbreviations: RAS, renin–angiotensin system; PTH, parathyroid hormone; TC, total cholesterol; TG, triglyceride; LDL, low-density lipoprotein; HDL, high-density lipoprotein; NO, nitric oxide; BNP, brain natriuretic peptide; hs-CRP, high-sensitive C-reactive protein; LV, left ventricle.

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
