# Peer review of "The Efficacy of Vitamins in the Prevention and Treatment of Cardiovascular Disease"

_ijms, 2024, doi:10.3390/ijms25189761_

Round 1

Reviewer 1 Report

Comments and Suggestions for Authors

Given the logic and coherence of this review article, the structure and organization of the paper are clear, and I recommend considering accepting this manuscript after revision. Here are some of my suggestions for the author.

In the introduction section of the article, the author mentions the classification and function of vitamins, but does not clearly explain the relationship between these vitamins and cardiovascular disease, resulting in the contents of the introduction section not being closely related to the topic of the article.

In terms of linguistic expression and writing style, the language of the essay should be more precise and concise, in line with the norms of academic writing. However, some sentences are not expressed smoothly enough and need some correction and improvement.

Some technical terms and abbreviations are used that may be difficult for non-expert readers to understand. Therefore, it is recommended that when professional terms and abbreviations are used, a corresponding explanation is provided or an appropriate explanation is given in the article so that the reader can better understand.

The conclusion is too long. Due to inconsistencies between experimental and clinical observations regarding the efficacy of vitamins in the treatment of cardiovascular disease, no firm conclusions can be drawn and it is important to explore the causes.  It is therefore suggested that a chapter should be added, devoted to a detailed discussion, which would be preferable to placing it directly in the concluding section.

In addition, it is suggested that the authors list and summarize the important literature in each section, so that the time chain and development threads of the study will be more clear. At the same time, it is suggested that the author should further strengthen the expression and discussion of his own views on the basis of literature induction.

Of course, if we can further systematically beautify several mechanism graphs, we can further increase the scholarship and value of the article.

Comments on the Quality of English Language

Fine.

Author Response

  1. …contents of the introduction section not closely related to the topic of the article.

Response: In accordance to this Reviewer, we have edited the introduction section to clarify the topic of the review and thus have added lines 48-62, 67-75 and 81-93 to describe the nature of the review.

  1. …the language of the essay should be more precise and concise.

 Response: We have improved sentence structure for enhanced readability.

  1. Need explanation for the use of some technical terms and abbreviations for better understanding for a non-expert reader.

 Response: We thank the reviewer for this comment. We have now amended in the revised manuscript.

  1. The conclusion is too long.

 Response: We have shortened the conclusion (pages 11-12, lines 606-612) in the revised manuscript as per suggestion of the reviewer. We now also provide a new section, section 7, lines 555-575 entitled “Perspectives” for a discussion on some of the causes that may be associated with some of the inconsistencies observed between experimental and clinical investigations on pages 11-12 in the revised manuscript.

  1. If further systematically beautify several mechanism graphs

Response: We thank the reviewer for this option, but in our opinion, the figures presented in the review are concise and provide a summary of the appropriate information covered in the review.

Reviewer 2 Report

Comments and Suggestions for Authors

This manuscript describes the association of vitamin status with cardiovascular diseases.

The authors did not describe which databases they used to collect information.

In the result part, authors should include a table gathering the main findings from the literature. Please provide.

There are several typos.  

Comments on the Quality of English Language

There are several typos.  

Author Response

  1. The authors did not describe the databases used to collect information.

Response: In accordance with this reviewer, we have added how the information presented in the review was collected in lines 90-93, on page 2 of the revised manuscript.

  1. Authors should include a table gathering the main findings from the literature.

Response: Since this would be a duplication of information, which is already described in the review article, we have not tabulated the main findings from the literature.

  1. There are several typos.        

Response: We apologise for these errors, which have now been corrected in the revised manuscript.

Reviewer 3 Report

Comments and Suggestions for Authors

Major concern: 

    Vitamins are essential nutrients for humans. There have been many research reports and literature reviews on diseases or syndromes caused by vitamin deficiency. This manuscript conducts a series of discussions on the impact of vitamins on cardiovascular diseases. However, the positive effects of water-soluble vitamins such as vitamin B1 and vitamin B6 still need to be strengthened. In addition, this literature review should not only elaborate on the literature reports, but also propose new insight and point out the future research direction of vitamins on cardiovascular diseases. 

Author Response

  1. Section on vitamin B1 and B6 need to be strengthened.

Response: We thank the reviewer for this comment. Accordingly, we have added to the section on B vitamins additional information on page 3 (lines 123-139), continuing on page 4 (lines 140-142 and 193-194) and page 5 (lines 208-220 and 236-243) in the revised manuscript.

  1. …propose new insights and point out the future research direction.

Response: In accordance to this reviewer and to address these points, we have ended each section with new insight and future research direction. In addition, as per Reviewer 1, we have added a new section 7, lines 555-575, in the revised manuscript, entitled “Perspectives”. This section also discusses new insights and points out future research direction.

Round 2

Reviewer 3 Report

Comments and Suggestions for Authors

The manuscript is now suitable for publication in this journal.